# Chromatographic Detection of 8-Hydroxy-2′-Deoxyguanosine in Leukocytes of Asbestos Exposed Workers for Assessing Past and Recent Carcinogen Exposures

**DOI:** 10.3390/diagnostics10040239

**Published:** 2020-04-21

**Authors:** Filippo Cellai, Stefano Bonassi, Alfonso Cristaudo, Alessandra Bonotti, Monica Neri, Marcello Ceppi, Marco Bruzzone, Mirta Milić, Armelle Munnia, Marco Peluso

**Affiliations:** 1Cancer Risk Factor Branch, Regional Cancer Prevention Laboratory, ISPRO-Study, Prevention and Oncology Network Institute, 50139 Florence, Italy; f.cellai@ispro.toscana.it (F.C.); armellemunnia@gmail.com (A.M.); 2Clinical and Molecular Epidemiology, IRCCS San Raffaele Pisana, 00166 Rome, Italy; stefano.bonassi@sanraffaele.it (S.B.); monicaneri2008@gmail.com (M.N.); 3Department of Human Sciences and Quality of Life Promotion, San Raffaele University, 00166 Rome, Italy; 4Department of Translational Research and New Technologies in Medicine and Surgery, University of Pisa, 56010 Pisa, Italy; alfonso.cristaudo@med.unipi.it; 5Occupational Medicine Unit, University of Pisa, 56010 Pisa, Italy; abonotti@yahoo.it; 6Unit of Clinical Epidemiology, IRCCS Ospedale Policlinico San Martino, 16131 Genoa, Italy; marcello.ceppi@hsanmartino.it (M.C.); marco.bruzzone@hsanmartino.it (M.B.); 7Mutagenesis Unit, Institute for Medical Research and Occupational Health, 10000 Zagreb, Croatia; mirtamil@gmail.com

**Keywords:** asbestos, leukocytes, 8-oxodG

## Abstract

Asbestos fibers include a group of silicate minerals that occur in the environment and are widely employed in occupational settings. Asbestos exposure has been associated to various chronic diseases; such as pulmonary fibrosis; mesothelioma; and lung cancer; often characterized by a long period of latency. Underlying mechanisms that are behind the carcinogenic effect of asbestos have not been fully clarified. Therefore; we have conducted an epidemiological study to evaluate the relationship between 8-hydroxy-2′-deoxyguanosine (8-oxodG), one of the most reliable biomarkers of oxidative stress and oxidative DNA damage; and asbestos exposure in the peripheral blood of residents in Tuscany and Liguria regions; Italy; stratified by occupational exposure to this carcinogen. Levels of 8-oxodG were expressed such as relative adduct labeling (RAL); the frequency of 8-oxodG per 10^5^ deoxyguanosine was significantly higher among exposed workers with respect to the controls; i.e., 3.0 ± 0.2 Standard Error (SE) in asbestos workers versus a value of 1.3 ± 0.1 (SE) in unexposed controls (*p* < 0.001). When the relationship with occupational history was investigated; significant higher levels of 8-oxodG were measured in current and former asbestos workers vs. healthy controls; 3.1 ± 0.3 (SE) and 2.9 ± 0.2 (SE), respectively. After stratification for occupational history; a significant 194% excess of adducts was found in workers with 10 or more years of past asbestos exposure (*p* < 0.001). 8-oxodG can be used for medical surveillance programs of cohorts of workers with past and recent exposures to carcinogens for the identification of subjects requiring a more intense clinical surveillance.

## 1. Introduction

Asbestos consists of a group of fibrous silicate minerals that occur in environment and is produced by certain industrial processes [1]. Asbestos is commonly divided into serpentine and amphibole groups based on the fiber characteristics and has been widely used as a constituent in buildings, machines, transport vehicles, and consumer products. 

The study of Doll and Knox first demonstrated a significant excess of lung cancer in asbestos workers [2]. Asbestos exposure has since then been considered by a number of IARC Working Groups and has been classified as a human carcinogen (Group 1) [3]. Currently, asbestos exposure has been associated with asbestosis, bronchogenic carcinoma, and a rare and aggressive cancer, mesothelioma [4,5]. Asbestos-induced carcinogenesis has been associated to its ability to induce indirect DNA damage, DNA strand breakage and chromosomal abnormalities [6]. During inflammatory response to asbestos fibers and iron-containing asbestos fibers, macrophages produce high amounts of cytokines and reactive oxygen species (ROS) [6]. Higher ROS levels can induce breaks or base modifications in DNA, resulting in oxidative DNA damage, including 8-hydroxy-2′-deoxyguanosine (8-oxodG). 8-oxodG, if not fully repaired, can induce G:C to T:A transversions and G:C to A:T transitions [7], the same kind of damage induced by some asbestos fibers [8]. 8-oxodG is one of multiple products of DNA oxidation that can be easily quantified and is commonly considered a reliable indicator of environmental exposure, and carcinogenic and mutagenic process [9,10,11]. The formation of 8-oxodG, as a result of exposure to asbestos fibers, in experiments cells including pulmonary epithelial cells and pleural mesothelial cells, has been described in review papers [12,13]. Significant relationships between occupational asbestos exposure and biomarkers of DNA and chromosomal damage and micronucleus frequencies have been extensively reported [1,14,15,16,17,18,19]. 

Before the introduction of official banning on asbestos in 1992 [20], Italy was the largest asbestos producer and, therefore, the rates of mesothelioma are still dramatically high in urban areas with past asbestos cement plants. For instance, in the city of Casale Monferrato, the mean annual incidence of mesothelioma was ranging up to 51.2 and 20.2 per 100,000/year among men and women, respectively in 2009–2013 [21]. Therefore, a public health program to increase epidemiological surveillance of mesothelioma, the ReNaM (Registro Nazionale Mesoteliomi), has been established in Italy [22] to monitor the rates of asbestos-related diseases of workers with former asbestos exposure [20]. ReNaM is active in most Italian regions and provides routine digitalized data on about 340 million person-years. In ReNaM, the majority (about 70%) of mesothelioma cases can be attributed to asbestos fiber occupational exposure including shipbuilding and repair, railway carriages maintenance and repair, iron and steel industry, and petrochemical and oil refinery [22]. In summary, oxidative stress is most likely involved in asbestos-related diseases. The goal of our project was to conduct a cross-sectional study aimed at comparing the presence of 8-oxodG in the leukocytes of workers exposed to asbestos with unexposed healthy controls, taking into account occupational history. For this purpose, the level of 8-oxodG was analyzed by ^32^P-postlabeling [23]. Our main purpose was to develop biomarkers that can easy identify those subjects with past exposure to asbestos with the highest levels of DNA damage and improve the effectiveness of welfare system and prevention policies of the ReNaM. 

## 2. Material and Methods

### 2.1. Study Population 

In the cohort, the beginning of asbestos exposure of asbestos former workers was 1980 and the end of exposure was 1992, whereas the group of current exposed workers was exposed to asbestos since 1974. Most of the former exposed workers had worked in the chemistry, metal-mechanics and shipbuilding sectors as repairmen, equipment operators, carpenters and mechanics; conversely, current asbestos workers consisted of subjects who were employed in construction industry and residential building settings, which are considered at high risk of contamination by asbestos even after the bans (IARC 2012). Study volunteers were randomly selected among a group of asbestos workers, who were under a program of health surveillance at University Hospital of Pisa, Pisa, Tuscany Region, Italy [24]. The program consisted of complete clinical examinations including Diffusion Lung CO, spirometry, chest X-ray, and the collection of occupational history by questionnaire administered by a trained interviewer. Asbestos workers were contacted by occupational physicians of the University Hospital of Pisa. Eligibility criteria were as follows: (1) to be employed in companies entailing recent (at least for 1 year) or past exposure to asbestos fiber in Pisa province (for asbestos workers), and (2) to have no reported history of occupational asbestos exposures (for healthy controls). Asbestos workers with mesothelioma; cancer; immune disease; and other pathologies, such as asbestosis and silicosis, were excluded from the study. Healthy controls were selected at the Ospedale Policlinico San Martino of Genoa among blood donors by local health service in Liguria Region, Italy. Controls included only subjects without a history of occupational and/or environmental exposure to asbestos and/or other carcinogens. The study was conducted in accordance with the guidelines of the Helsinki Declaration and approved by the Institutional Review Board of the IRCCS San Raffaele, Rome, Italy (23 April 2012, Prot. N.14/12). All the participants were informed about the aims of the study and provided a written informed consent. Details about age, gender, professions, life-style habit, and occupational status and history were obtained by means of a detailed questionnaire administered by personnel experienced in industrial hygiene.

### 2.2. DNA Extraction and Purification 

DNA was extracted and purified from buffy coat with organic solvents after RNase A and T_1_ and proteinase K treatments [25,26]. DNA concentration and purity were evaluated by spectrophotometer. Coded samples were stored at −80 °C. Before chromatographic analysis, DNA (2 μg) was hydrolyzed by micrococcal nuclease (21.45 mU/μL) and spleen phosphodiesterase (6.0 mU/μL) at 37 °C for 4.5 h [27]. 

### 2.3. ^32^P-DNA Postlabeling 

Diluted digest (20 ng/µL) was treated with 10 μCi of carrier-free [γ-^32^P]ATP (3000 Ci/mM) and 2.0 U of polynucleotide kinase T4 (10 U/μL) at 37 °C for 45 min. ^32^P-labeled digests were treated with 1.2 U of nuclease P1 (1.9 U/μL) at 37 °C for 60 min. ^32^P-labeled samples were examined by polyethyleneimine cellulose thin-layer chromatography plates [23]. An aliquot of 2.5 µL of nuclease P1-treated ^32^P-labeled digests was spotted at chromatogram origin and developed overnight into a 3-cm-long Whatman 1 paper wick with 1.5 M formic acid for the first dimensional chromatography. For the second dimensional direction, sheets were developed at the right angle to the previous course with 0.6 M ammonium formate, pH 6.0. Then, 8-oxodG detection and quantification were obtained by storage phosphor imaging technique with intensifying screens from Molecular Dynamics. The screens were scanned using a Typhoon 9210. After background subtraction, levels of 8-oxodG were expressed such as relative adduct labeling (RAL) = screen pixel in 8-oxodG spot/screen pixel in total nucleotides. Levels of 8-oxodG were corrected across experiments on the recovery of internal standard.

### 2.4. Statistical Analysis 

Levels of 8-oxodG were expressed per 10^5^ deoxyguanosine (dG). Data were log transformed to stabilize the variance and normalize the distribution. Log-normal regression models, with age (years), smoking habit (non-smokers, ex-smokers and smokers), occupational status (asbestos worker vs. unexposed controls) and occupational history (years) as independent variables, were used to study the association between asbestos exposure and 8-oxodG. Asbestos workers were then sub-grouped according to current and former asbestos exposures in two groups: (1) workers with current potential exposure to asbestos, and (2) workers with past exposure to asbestos. Duration of employment entailing exposure to asbestos fibers was then used as a proxy for length of carcinogen exposure at workplace. MR estimates and its 95% C.I. were employed as a measure of effect [28] for each level of the predictor variables relative to unexposed controls. Statistical analysis was done using the software Stata/SE 13.1 (StataCorp LP, College Station, TX, USA).

## 3. Results

### 3.1. Demographic Variables 

The present cross-sectional study involved 185 male subjects, 54.1 ± 8.0 (SD) years of age. There were 98 asbestos workers, 53.5 ± 9.7 (SD) years of age, with a mean number of years of asbestos fiber exposure of 8.5 ± 6.4 (SD), 31% smokers; and 87 unexposed controls, 54.9 ± 5.5 (SD) years of age, 18% smokers. All participants were males, because most of the asbestos workers are males. Asbestos workers were then classified by occupational history, e.g., former and current asbestos exposure. After stratification, there were 39 current asbestos workers, 45.3 ± 8.4 (SD) years of age, and 59 former asbestos workers, 58.9 ± 6.1 (SD) years of age. Frequency of clinical examinations was comparable among the different groups of exposed workers. Control subjects did not receive specific examinations but the admission protocol for blood donors includes major respiratory diseases. Since no direct data about f/mL years and/or type of asbestos were available, the intensity of exposure was estimated through the duration and the characteristics of various occupations 

### 3.2. 8-OxodG and Asbestos Workers

In this study, we examined the frequency of 8-oxodG in the leukocytes of the study participants using ^32^P-postlabeling [23] to evaluate genotoxic effects induced by occupational exposure to asbestos fibers. The specific analysis of 8-oxodG was performed by a chromatographic system known to be effective for the detection and the quantitative examination of this kind of adduct [29,30,31,32,33,34]. Chromatographic findings show that a typical 8-oxodG spot pattern was observed in chromatographic sheets of study population by ^32^P-postlabeling (Figure 1). The intensity of 8-oxodG spots was commonly stronger in the plates of asbestos and workers when compared to unexposed controls (Figure 1). 

The qualitative analysis of autoradiographic profiles showed that heavier patterns of 8-oxodG spots were detected in the chromatographic plates of current and former exposed asbestos workers as compared to healthy controls (Figure 2). The average frequencies of 8-oxodG, expressed per 10^5^ dG, according to demographic and life-style habits, are reported In Table 1. The overall mean levels of 8-oxodG in the study participants were of 2.2 ± 0.1 Standard Error (SE). Results indicate that values tended to be inversely associated to educational level. Indeed, lower adduct frequencies of 8-oxodG were found in the volunteers with the highest education level in respect to those with the primary education level alone, 1.3 ± 0.2 (SE) vs. 2.3 ± 0.2 (SE), respectively. A significant MR value of 0.65 was also computed (95% C.I., 0.48–0.88). Conversely, adduct formation was not significantly associated with smoking habits (MR = 1.09, 95% C.I., 0.74–1.60).

In Table 2, the mean amounts of 8-oxodG, expressed per 10^5^ dG, according to occupational exposure status are reported. An increased adduct frequency was detected in asbestos workers as compared to unexposed controls, 3.0 ± 0.2 (SE) vs. 1.3 ± 0.1 (SE), respectively. The multivariate analysis found a 136% excess of 8-oxodG in asbestos workers relative to controls, after adjusting for age, educational level and smoking habits (95% C.I. 1.99–2.81). When the relationship with occupational history was investigated, higher levels of 8-oxodG were measured in current and former asbestos workers vs. healthy controls, 3.1 ± 0.3 (SE) and 2.9 ± 0.2 (SE), respectively. The statistical analysis shows that significant 143% and 132% increments of oxidative adducts were present in both the occupational groups (MR = 2.43, 95% C.I. 1.90–3.12 and MR = 2.32, 95% C.I. 1.88–2.86, respectively). Moreover, the highest frequency of 8-oxodG was detected in those subjects who have worked for 10 or more years as compared to controls, 3.5 ± 0.5 (SE) vs. 1.3 ± 0.1 (SE), respectively. Interestingly, the highest adduct excess of 194% (MR = 2.94; 95% C.I. 2.11–4.11) was found in this sub-group (Table 2), after adjusting for the confounding factors. Subsequently, we have analyzed the relationship of adducts with occupation duration in the group of workers alone. A 34% increment of adducts (MR = 1.34), although not statistically significant, persisted among those with a ≥10 years exposure length.

## 4. Discussion

Exposure to asbestos fibers still represents a main occupational risk in Europe, despite the prohibition of extract, manufacturing and process asbestos products in 1976 (directive 2003/18/EC), and the subsequent ban in 2005 [35]. Asbestos fiber exposure can lead to chronic lung inflammation and to an aggressive cancer like the mesothelioma [4,5]. Notably, fiber dimension and surface properties are determinants of asbestos carcinogenicity, however, despite the well-known toxicity of this carcinogen, the underlying mechanisms by which asbestos causes chronic diseases are still incompletely understood [35]. Previously, we have conducted a cross-sectional study to analyze the relationship between 3-(2-deoxy-β-d-erythro-pentafuranosyl)pyrimido[1,2-α]purin-10(3H)-one deoxyguanosine (M_1_dG) adducts, a biomarker of oxidative stress and peroxidation of lipids in cell membranes [36], and asbestos fiber exposure in a cohort of workers living in Tuscany [37]. In that study, we demonstrated a significant association between occupational asbestos exposure and the generation of lipid peroxidation-related DNA adducts in asbestos workers, who were employed in several industries in Tuscany region, as shown by greater levels of exocyclic M_1_dG in the leukocytes of asbestos workers as compared to unexposed controls. In the current study, we asked whether asbestos workers experienced also enhanced frequency of 8-oxodG, a biomarker of oxidative stress and cancer risk [9,10,11], that could indicate increased health risks in later life. Indeed, biomarkers, such as the measurement of oxidative DNA damage, can be used to evaluate the genotoxic effects of volatile carcinogens in various occupational settings [38,39,40,41].

We report here, further evidence supporting oxidative stress hypothesis for the damaging actions of asbestos fibers [42], which suggests that phagocytic cells produce large amounts of reactive oxygen species, due to their inability to digest asbestos fiber. A main finding shows that occupational exposure to asbestos fibers causes the formation of high amount of oxidative DNA adducts in the peripheral leukocytes of asbestos workers. A significant 136% excess of 8-oxodG occurred in the asbestos workers as compared to controls (MR = 2.36; 95% C.I. 1.99–2.81). Then, we examined the generation of oxidative damage according to occupational history. In this case, subjects were classified as currently exposed asbestos workers especially employed in the construction industry, i.e., a kind of occupation that can still imply asbestos exposure [35]. Conversely, former asbestos workers were especially from mechanic, naval, petrochemical industries, and pottery and ceramic plants. Our investigation clearly shows that past occupational exposure to asbestos appears to be still a relevant source of genetic damage in the peripheral blood of workers as well as current workers. Multivariate statistical analysis showed that large excesses of adduct were still detectable after stratification for occupational history in current and past asbestos workers (MR = 2.43, 95% C.I. 1.90–3.12 and MR = 2.32, 95% C.I. 1.88–2.86, respectively).

The most remarkable finding of the study was the increased adduct frequency in the group of formerly exposed workers. This data has a special value since highly increased amounts of 8-oxo-dG increase the probability of mutagenic G:T-transversions during cell division, which are frequently found in tumor-relevant genes. Additionally, it appears the workers exposed to high levels of asbestos fibers in the past, have greater amounts of promutagenic lesions that can still be recognized, and consequently reflect the potential risk of occupational-related cancers. These results support the use of such adducts in leukocytes of workers for the early detection of susceptible at-risk individuals in cohort of workers, as well as a reliable biomonitoring of such cohort of subjects with past exposures to asbestos fibers.

Adduct results have a great importance since adducts levels have been associated with higher risks of developing lung cancer [43]. Previous investigations showed increased amounts of different types of DNA damage in asbestos workers using various biomolecular approaches [1,14,15,16,17,18,19]. A high frequency of 8-oxodG was detected in peripheral blood [1,17] or urine of asbestos workers [18,19]. Particularly, Marczynski et al. [17] performed an investigation on a larger cohort of asbestos workers that evaluated the relationship of asbestos with 8-oxodG. In that study, a significant increase in adduct levels was observed in the white blood cells of asbestos workers relative to controls, and, similarly, a lack of correlation was observed with age and smoking habits. However, no association with duration of asbestos exposure was found although more detailed information on asbestos fibers exposure types was collected. High frequency of micronucleus was also reported in subjects who were diagnosed with pleural plaques due to asbestos exposure [14,16]. Recently, DNA damage in buccal cells was investigated in Italian study groups with different types of exposure to asbestos fibers, i.e., none, past, or current [44]. In this study, significantly higher micronuclei levels, as well as other nuclear alterations, were found in the group with past asbestos exposure compared with the controls, whereas lower levels of micronuclei were found in the group with current exposure.

The mechanisms underlying 8-oxodG production by this carcinogen in asbestos workers can be due to the accumulation of fibers on pleural surface and by the interaction with the mesothelial cell layer, which can lead to the formation of free radicals. High ROS levels can result from chronic inflammation generated by the prolonged phagocytic activity of macrophages while eliminating persistent fibers [19]. The presence of catalytic iron on the surface of asbestos fibers can be also a main source of ROS production. For instance, amphiboles have a higher iron concentration than serpentines, which are more mutagenic. This difference in iron level in harm potential can be due to the variation in the activity of the superficial iron. Fenton-type reactions catalyzed by existing iron on the surface of asbestos fibers [45] can induce free radicals, which can diffuse to peripheral blood cells through lung microvascular endothelium [46]. ROS generated by persistent inflammation and macrophages and neutrophils activation in lung parenchyma can indirectly cause 8-oxodG and other effects such as lipid peroxidation, and subsequently M_1_dG, in addition to activation of signaling pathways and cell proliferation [35]. For instance, chrysotile asbestos-induced autophagy, that can be partially mediated by ROS and TP53 related pathways [47], has been correlated to altered signaling through AKT/mTOR and c-Jun N-terminal kinase signaling pathways [48]. Multiple mechanisms by which chrysotile asbestos fibers induce pulmonary disease have been identified [49]; however, ROS and inflammatory cytokines caused by pleural accumulation of asbestos fiber exposures seem to be associated to various kind of DNA damage, which can induce oncogene activation, cell proliferation and higher susceptibility to mutations.

Asbestos-related diseases can appear after a long period of latency, e.g., 10–20 years for lung cancer, and up to 50 years for pleural malignant mesothelioma [35]. The main finding of the current study clearly shows that past occupational exposure to this carcinogen, even several years after asbestos exposure has ceased, appears to be still a relevant source of genetic damage in the peripheral blood of workers. A strong increase of 8-oxodG was observed in workers who have been employed for 10 or more years in asbestos plants as compared to unexposed controls. The generation of oxidative adducts may reflect the concentrations of asbestos fibers that have been accumulated in lung parenchyma, where those fibers can cause chronic inflammation, leading in turn to greater ROS production, which may overwhelm intrinsic anti-oxidant defense. The burden of persistent fibers in the lung appears to play a role in increasing the levels of 8-oxodG in past asbestos workers.

Despite the heterogeneity of working conditions, we have noted that workers no longer exposed, but who require constant medical observation for higher risk of asbestos related chronic diseases at Pisa Hospital, and currently exposed asbestos workers have comparable amounts of 8-oxodG. A possible reason for high adduct levels in this category could be the inappropriate use of safety devices, but also the necessity of revision and improvement of these equipment. Nevertheless, our population was employed in different settings, such as mechanic, naval, petrochemical, and building industries, and pottery and ceramic plants, thereby, workers could have also experienced exposures to other concomitant volatile carcinogens. Elevated 8-oxodG levels can reflect concomitant exposures to other volatile carcinogens in the workplaces, including silica dust, heavy metals, formaldehyde, aromatic amines, and polycyclic aromatic hydrocarbons, as well as secondary DNA damage produced from by-products of primary DNA damage or lipid membrane peroxidation. An example of this relationship comes from previous epidemiological studies measuring oxidative damage in residents living nearby a large petrochemical complex in Sardinia island, Italy [50] and among workers exposed to silica dust in Tuscany, Italy [39].

The potential role of smoking habits, age and educational levels as potential confounders of 8-oxodG levels was carefully considered in our study. Age and cigarette smoking have been associated to ROS generation, and an increased level of oxidative DNA damage could be expected [51,52]; nevertheless, these factors had a limited influence on the amounts of 8-oxodG. This result is not surprising, since various occupational studies have not reported higher levels of 8-oxodG in smokers [53]. The high prevalence of light smokers in our cohort could account for this lack of association, alternatively this could be the rise of 8-oxodG, which is short lived in transient exposure. In fact, after repeated exposures, up-regulation of DNA repair could counteract an increased rate of 8-oxodG formation leaving the steady state level of 8-oxodG in DNA unchanged [54]. Also in the case of age-related genotoxic effects, it appears that rates of oxidative DNA damage alterations associated to aging are difficult to consistently define [55]. Instead, educational levels, a parameter that can reflect the socioeconomic status, was found to be a significant risk factor for higher 8-oxodG levels. An inverse association of adducts with high-school education levels has already been reported [56]. The observed relationship between high school education and 8-oxodG levels could be due to the fact that subjects who have lower educational levels live in more polluted areas or work in occupations characterized by higher exposures to environmental carcinogens [56] or alternatively to uncontrolled exposure environment.

Accurate measurement of 8-oxodG in leukocyte DNA is desirable in toxicology because it can improve the knowledge of how oxidative stress, and more specifically oxidative DNA damage, can induce adverse health effects. Notably, the validation of DNA alterations for their potential use as biomarkers of exposure in biomonitoring studies should require establishing causal relationships with occupational status in cross-sectional studies. Herein, we have used ^32^P-postlabeling chromatography, which is an analytical system offering high sensitivity [23,32], over high-performance liquid chromatography with on-line electrochemical detection (HPLC–EC) or gas chromatography, followed by on-line mass spectrometry (GC–MS), methods which have their typical advantages and disadvantages [57]. In respect to HPLC–EC, ^32^P-postlabeling 8-oxodG chromatography presents various similarities, e.g., high sensitivity and enzymatic digestion of DNA, together with important methodological differences, e.g., amounts of DNA used and nuclease P1 treatment. Additionally, this technique has been frequently used to analyze the generation of 8-oxodG in experimental animals and human biomonitoring studies [58,59,60] and in a pancreatic cancer case-control study [23]. Chromatographic methods, such as HPLC and GC-MS, are generally available in clinical analysis in both routine and research laboratories for the study of various analytics such as drugs, hormones, metabolomics, lipidomics, and volatile organic compounds. Whereas, 8-oxodG chromatography, HPLC coupled with EC or to electrospray tandem mass spectrometry (HPLC-MS/MS), and liquid chromatography (LC-MS/MS)-based methodologies for analysis of 8-oxo-dG, are less frequent in routine clinical settings, but the important features of these applications should be emphasized. For instance, 8-oxodG chromatography appears to be a good method for the detection of one of the predominant forms of free radical-induced oxidative lesions, commonly used as a biomarker for oxidative stress, lipid peroxidation-related DNA adducts and cancer risk or carcinogenesis.

## 5. Conclusions

The analysis of 8-oxodG in asbestos workers can contribute to identify subjects at increased level of genotoxic damage, a condition that in turn implies a higher risk of lung cancer and other chronic diseases. Adduct measurement could be used in combination with other complementary biomarkers, such as epigenetic alterations [61,62]. This approach can be used for medical surveillance programs of cohorts of workers with past exposures to carcinogens and for the identification of subjects at higher risk of cancer. In this case, adducts can identify subgroups at higher risk, which will require a more intense clinical surveillance, or can be further studied for a possible individual risk assessment. Our results endorse policy initiatives aimed at detecting and eliminating exposure to asbestos fibers and to prevent potential health hazards in occupational settings.

## Figures and Tables

**Figure 1 diagnostics-10-00239-f001:**
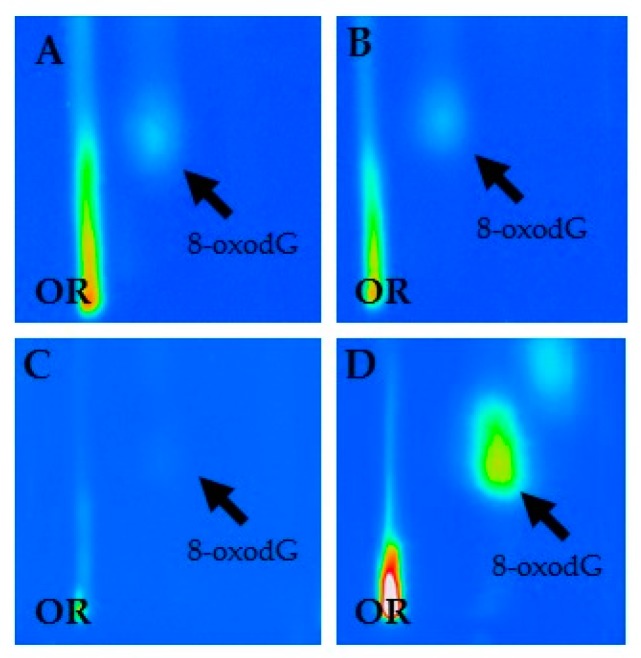
Characteristic chromatographic patterns of 8-hydroxy-2′-deoxyguanosine (8-oxodG) in current (**A**) and former (**B**) exposed asbestos workers, unexposed healthy controls (**C**) and positive standard (**D**).

**Figure 2 diagnostics-10-00239-f002:**
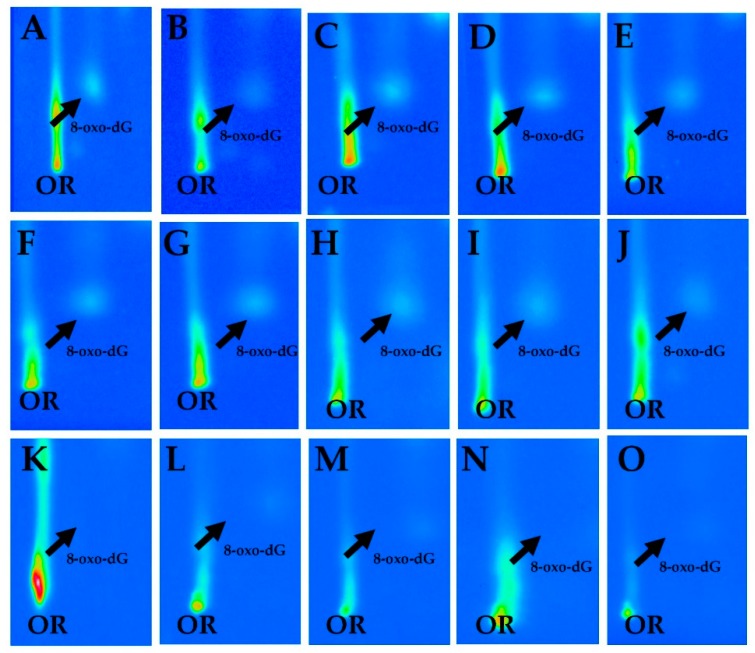
Panel of representative chromatographic patterns of 8-hydroxy-2′-deoxyguanosine (8-oxodG) spots in five current (**A**–**E**) and five former exposed asbestos workers (**F**–**J**), and five unexposed healthy controls (**K**–**O**).

**Table 1 diagnostics-10-00239-t001:** Average frequency of 8-hydroxy-2′-deoxyguanosine (8-oxodG) in study population by age, educational level and smoking habits. Adjusted estimated of risk are expressed as Mean Ratio (MR) with their respective 95% confidence interval (95% CI).

		8-OxodG per 10^5^ Deoxyguanosine (dG)		
	N ^a^	Mean ± Standard Error (SE)	MR, 95% C.I.	*p*-Values ^b^
Age (years)				
≤52	67	2.3 ± 0.2	Reference	
53–58	63	2.1 ± 0.2	1.00 (0.77–1.29)	0.978
≥59	55	2.1 ± 0.2	0.89 (0.68–1.17)	0.408
Educational level				
Primary education	86	2.3 ± 0.2	Reference	
Secondary education	72	2.3 ± 0.2	0.84 (0.67–1.06)	0.139
University	27	1.3 ± 0.2	0.65 (0.48–0.88)	0.006
Smoking status				
Non-smokers	72	2.2 ± 0.2	Reference	
Ex-smokers	66	2.0 ± 0.2	0.98 (0.67–1.43)	0.918
Smokers	46	2.4 ± 0.3	1.09 (0.74–1.60)	0.668

^a^ Some figures do not add up to total due to missing values. ^b^
*p*-values (Test of Wald) were adjusted for confounding factors.

**Table 2 diagnostics-10-00239-t002:** Mean levels of 8-hydroxy-2′-deoxyguanosine (8-oxodG) in study subjects considering asbestos exposure, occupational history and duration.

	8-OxodG per 10^5^ Deoxyguanosine (dG)			
	N ^a^	Mean ± Standard Error (SE)	MR, 95% C.I.	*p*-Values ^b^
Asbestos exposure				
Unexposed controls	87	1.3 ± 0.1	Reference	
Asbestos workers	98	3.0 ± 0.2	2.36 (1.99–2.81)	<0.001
Current and past exposure				
Unexposed controls	87	1.3 ± 0.1	Reference	
Former asbestos workers	59	2.9 ± 0.2	2.32 (1.88–2.86)	<0.001
Current asbestos workers	39	3.1 ± 0.3	2.43 (1.90–3.12)	<0.001
≤4 years asbestos workers	18	2.8 ± 0.4	2.21 (1.64–2.99)	<0.001
5–9 years asbestos workers	21	2.9 ± 0.3	2.40 (1.79–3.23)	<0.001
≥10 years asbestos workers	16	3.5 ± 0.5	2.94 (2.11–4.11)	<0.001
Occupational history				
≤4 years	18	2.8 ± 0.4	Reference	
5–9 years	21	2.9 ± 0.3	1.09 (0.73–1.64)	0.672
≥10 years	16	3.5 ± 0.5	1.34 (0.85–2.10)	0.197

^a^ Some figures do not add up to total due to missing values. ^b^
*p*-values (Test of Wald) were adjusted for confounding factors.

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
