# Peer review of "Chromatographic Detection of 8-Hydroxy-2′-Deoxyguanosine in Leukocytes of Asbestos Exposed Workers for Assessing Past and Recent Carcinogen Exposures"

_diagnostics, 2020, doi:10.3390/diagnostics10040239_

Round 1
Reviewer 1 Report
This study describes raised levels of 8-hydroxy-2'deoxyguanosine as a marker of oxidative stress in the peripheral blood of workers exposed to asbestos. A It would be good to know more about the cohort, i.e. when was the last exposure- what is meant by 'recent' exposure since a ban was declared 1992? An attempt to quantify exposure by f/mL years and/or type of asbestos would be useful. I understand the study population size may be too small for further subdivision, but the data should be at least be there? Where patients with asbestosis, lung cancer, mesothelioma excluded? The smoking result is curious- given that smoking is also demonstrated to act via ROS generation at least in current smokers an increased level may be expected, -- discussion about this finding would be good- is this thought to be because rise of 8IHdG short lived in transient exposure or...? Also, levels are generally described as higher in older subjects. This is not seen in this cohort- could the authors please comment> What was the rationale for using the chromatography method over the HPLC methods that others have used, and how do the findings compare? How available in a routine clinical lab is this assay compared to HPLC, which seems to be used more commonly? The study design of the Marczynski study (reference 23 in the paper) is very similar. Like the current study, Macrzynski et al looked at white blood cells (leukocytes), but the cohort was larger and there was more information on exposure types (e.g. fibre type). How did the methods (including analytical metgod) used in the current study compare?Author Response
Reviewer 1
This study describes raised levels of 8-hydroxy-2'deoxyguanosine as a marker of oxidative stress in the peripheral blood of workers exposed to asbestos.
Question N.1.It would be good to know more about the cohort, i.e. when was the last exposure- what is meant by 'recent' exposure since a ban was declared 1992?
Answer. In the group of asbestos former workers exposure started in the 1980 and ended in the 1992 after the ban or with retirement. These workers had worked in the chemistry, metal-mechanics and shipbuilding sectors as repairmen, equipment operators, carpenters and mechanics. The group of current exposed workers consisted of subjects who were employed in construction industry and residential building settings, both activities are considered at high risk of contamination by asbestos even after the bans (IARC 2012).
This paragraph was added to 2.1 Study population section.
Question N.2. An attempt to quantify exposure by f/mL years and/or type of asbestos would be useful. I understand the study population size may be too small for further subdivision, but the data should be at least be there?
Answer. We agree with the reviewer about the importance of exposure assessment. Our consideration of this issue is now reported in the revised version of the manuscript. Since no direct data about f/mL years and/or type of asbestos were available the intensity of exposure was estimated through the duration and the characteristics of various occupations.
This comment was added to 3.1. Demographic variables section.
Question N.3.Where patients with asbestosis, lung cancer, mesothelioma excluded?
Answer. The presence of mesothelioma, cancer, and immune disease was an exclusion criteria of the study. Before statistical analysis also workers with other pathologies, i.e., 7 asbestos workers with asbestosis and 1 asbestos workers with silicosis, were excluded.
This paragraph was added to 2.1 Study population section. Data on Tables and text were also accordingly modified.
Question N.4.The smoking result is curious- given that smoking is also demonstrated to act via ROS generation at least in current smokers an increased level may be expected, -- discussion about this finding would be good- is this thought to be because rise of 8IHdG short lived in transient exposure or...?
Answer. The potential role of smoking habit, age and educational levels as potential confounders of on 8-oxodG levels was carefully considered in our study. Age and cigarette smoking have been associated to ROS generation and an increased levels of oxidative DNA damage could be expected (Lodovici et al. 2005; Nie et al. 2013), nevertheless, these factors had a limited influence on the amounts of 8-oxodG. These result is not surprising, since various occupational studies have not reported higher levels of 8-oxodG in smokers (Pilger and Radiger 2006). The high prevalence of light smokers in our cohort could explain this lack of association, alternatively this could be due to the rise of 8-oxodG short lived in transient exposure. In fact, after repeated exposures, up-regulation of DNA repair could counteract an increased rate of 8-oxodG formation leaving the steady state level of 8-oxodG in DNA unchanged (Risom et al. 2003). Also in the case of for age related genotoxic effects, it appears that rates of oxidative DNA damage alterations associated to aging are difficult to consistently define (Jacob et al. 2013). Instead, educational levels, a parameter that can reflect the socioeconomic status, was found to be a significant risk factor for higher 8-oxodG levels. An inverse associationof adducts with high-school education levels has been already reported (Rundle et al. 2012).The observed relationship between high school education and 8-oxodG levels be due to the fact that subjects who have lower educational levels live in more polluted areas or workin occupations with characterized by higher exposures to environmental carcinogens (Rundle et al. 2012) or alternatively to uncontrolled exposure environment.
This paragraph was added to 4.1 Discussion section.
Question N.5.Also, levels are generally described as higher in older subjects. This is not seen in this cohort- could the authors please comment>
Answer.
see the answer to the Question N.5.
Question N.6. What was the rationale for using the chromatography method over the HPLC methods that others have used, and how do the findings compare?
Answer. Accurate measurement of 8-oxodG in leukocyte DNA is desirable in toxicology because since it can improve the knowledge of how oxidative stress, and more specifically oxidative DNA damage, can induce adverse health effects. Notably, the validation of DNA alterations for their potential use as biomarkers of exposure in biomonitoring studies should require establishing causal relationships with occupational status in cross-sectional studies. Herein, we have used 32P-postlabeling chromatography, is an analytical system offering high sensitivity (Balansky et al. 2014; Mohamadkhani et al. 2017), over high-performance liquid chromatography with on-line electrochemical detection (HPLC–EC) or gas chromatography followed by on-line mass spectrometry (GC–MS), methods which have their typical advantages and disadvantages(Korkmaz et al. 2018). In respect to HPLC–EC,32P-postlabeling 8-oxodG chromatography presents various similarities, e.g., high sensitivity and enzymatic digestion of DNA, together with methodological differences, e.g.,amounts of DNA used and nuclease P1 treatment. Additionally, this techniquehas been also widely used to analyze the generation of 8-oxodG in experimental animals and in human biomonitoring studies (Bolognesi et al. 2016; Izzotti et al. 2020; La Maestra et al. 2015) and in a pancreatic cancer case-control study (Mohamadkhani et al. 2017).
This paragraph was added to 4.1 Discussion section.
Question N.7. How available in a routine clinical lab is this assay compared to HPLC, which seems to be used more commonly?
Answer. Chromatographic methods, such as HPLC, and GC-MS, are generally available in clinical analysis in both routine and research laboratories to the study of various analytics such as drugs, hormones, metabolomics, lipidomics, and volatile organic compounds.Whereas, 8-oxodG chromatography, HPLCcoupled with to EC or to electrospray tandem mass spectrometry (HPLC-MS/MS), and liquid chromatography (LC-MS/MS)based methodologies for analysis of 8-oxo-dG are less frequent in routine clinical settings, but the important features of these applications should be emphasized.For instance, 8-oxodG chromatography appears to be a good method for the detection of one of the predominant forms of free radical-induced oxidative lesions, commonly used as a biomarker for oxidative stress, lipid peroxidation-related DNA adducts and cancer risk or carcinogenesis.
This paragraph was added to 4.1 Discussion section.
Question N.8. The study design of the Marczynski study (reference 23 in the paper) is very similar. Like the current study, Macrzynski et al looked at white blood cells (leukocytes), but the cohort was larger and there was more information on exposure types (e.g. fibre type).
Answer. Marczynski et al. (Marczynski et al. 2000) have performed an investigation on a larger cohort of asbestos workers to evaluated the relationship of asbestos with 8‐oxodG. In that study, a significant increase in adduct levels was observed in the white blood cells of asbestos workers relative to controls, and, similarly, a lack of correlation wasobserved with age, and smoking habit. However, none association with duration of asbestos exposure was found although there more detailed information on asbestos fibers exposure types was collected.
This paragraph was added to 4.1 Discussion section.
Question N.9. How did the methods (including analytical metgod) used in the current study compare?
Answer.
see the answer to the Question N.6.
Reviewer 2 Report
The manuscript is easy to read and the study design is well defined.
It is relevant that the present study clearly shows that past occupational exposure to asbestos appears to be still a relevant source of genetic damage in the peripheral blood of workers as well as current workers.
Methods are accurate and appropriate as 8-oxodG chromatography is a good method for the detection of one of the predominant forms of free radical-induced oxidative lesions, widely used as a biomarker for oxidative stress, lipid peroxidation-related DNA adducts and cancer risk or carcinogenesis.
Minor concerns:
- in my opinion the title is not so appropriate to the findings of the study;
- please, insert the number for approved document by the Institutional Review Board
- DNA extraction and purification Section should be described in details or some references should be added;
- the authors should specify better in the text or in the legends the meaning for "dG" and "SE";
- Figure 1. The authors should make a panel showing at least 15 current and 15 former exposed asbestos workers and 15 unexposed healthy controls chromatographic patterns;
- please, provide a table with clinical examinations, if different, in the three different groups;
- none of the asbestos workers have cancer or immune disease or other pathologies. It should be clearly written;
- please, the authors should discuss the data concerning the years of exposure for asbestos workers;
- it is a little bit not understandable that "lower adduct frequencies of 8-oxodG were found in the volunteers with the highest education level in respect to those with the primary education level alone, 1.3 ± 0.2 (SE) vs. 2.3 ± 0.3 (SE), respectively". What about the impact of the uncontrolled exposure environment and of the education levels? Please, discuss them.
Author Response
Reviewer 2
The manuscript is easy to read and the study design is well defined.
It is relevant that the present study clearly shows that past occupational exposure to asbestos appears to be still a relevant source of genetic damage in the peripheral blood of workers as well as current workers.
Methods are accurate and appropriate as 8-oxodG chromatography is a good method for the detection of one of the predominant forms of free radical-induced oxidative lesions, widely used as a biomarker for oxidative stress, lipid peroxidation-related DNA adducts and cancer risk or carcinogenesis.
Minor concerns:
Question N.1.in my opinion the title is not so appropriate to the findings of the study;
Answer.
The title was modified as suggested. The new title is the following: “Chromatographic detection of 8-hydroxy-2'-deoxyguanosine in leukocytes of asbestos exposed workers for assessing past and recent carcinogen exposures”
Question N.2.please, insert the number for approved document by the Institutional Review Board
Answer.
The study was conducted in accordance with the guidelines of the Helsinki Declaration, and approved by the Institutional Review Board of the IRCCS San Raffaele, Rome, Italy (Prot. N.14/12)
This information was added to 2.1 Study population section.
Question N.3.DNA extraction and purification Section should be described in details or some references should be added;
Answer.
Two references were added:
- Godschalk R, Remels A, Hoogendoorn C, van Benthem J, Luijten M, Duale N, Brunborg G, Olsen A-K, Bouwman FG, Munnia A and others. 2018. Paternal Exposure to Environmental Chemical Stress Affects Male Offspringns Hepatic Mitochondria. ToxicologicalSciences 162(1):241-250.
- Zanoni TB, Hudari F, Munnia A, Peluso M, Godschalk RW, Zanoni MV, denHartog GJ, Bast A, Barros SB, Maria-Engler SS and others. The oxidation of p-phenylenediamine, an ingredient used for permanent hair dyeing purposes, leads to the formation of hydroxyl radicals: Oxidative stress and DNA damage in human immortalized keratinocytes. ToxicolLett 239(3):194-204.
These references were added to 2.2 DNA extraction and purification section.
Question N.4. the authors should specify better in the text or in the legends the meaning for "dG" and "SE";
Answer.
The meaning of dG (deoxyguanosine) and SE (Standard Error) is now specified in the text and in the corresponding legends.
Question N.5.Figure 1. The authors should make a panel showing at least 15 current and 15 former exposed asbestos workers and 15 unexposed healthy controls chromatographic patterns;
Answer.
A new Figure 2 with a panel of representative chromatographic patterns of 8-hydroxy-2′-deoxyguanosine (8-oxodG) spots in five current (A-E) and five former exposed asbestos workers (F-J), and five unexposed healthy controls (K-O) was added to 3.2. 8-oxodG and asbestos workers section.
Question N.6.please, provide a table with clinical examinations, if different, in the three different groups;
Answer.
Frequency of clinical examinations was comparable among the different groups of exposed workers. Control subjects did not receive specific examinations but the admission protocol for blood donors includes major respiratory diseases.
This information was added to 3.1. Demographic variables section.
Question N.7. none of the asbestos workers have cancer or immune disease or other pathologies. It should be clearly written;
Answer.
see the answer to the Question N.3 of the first reviewer.
Question N.8.please, the authors should discuss the data concerning the years of exposure for asbestos workers;
Answer.
The most remarkable finding of the study was the increased adduct frequency in the group of formerly exposed workers. This data has a special value since highly increased amounts of 8-oxo-dG increases the probability of mutagenic G to T-transversions during cell division, which are frequently found in tumor-relevant genes. Additionally, it appears the workers exposed to high levels of asbestos fibers in the past, have greater amounts of promutagenic lesions that can be still recognized, and the consequently reflect the potential risk of occupationally related cancers. These results support the use of such adducts in leukocytes of workers for the early detection of susceptible at-risk individuals in cohort of workers, as well as a reliable biomonitoring of such cohort of subjects with past exposures to asbestos fibers.
This paragraph was added to 4.1 Discussion section.
Question N.9.it is a little bit not understandable that "lower adduct frequencies of 8-oxodG were found in the volunteers with the highest education level in respect to those with the primary education level alone, 1.3 ± 0.2 (SE) vs. 2.3 ± 0.3 (SE), respectively". What about the impact of the uncontrolled exposure environment and of the education levels? Please, discuss them.
Answer.
see the answer to the Question N.4 of the first reviewer.